# Recipes for Unbiased Reward Modeling Learning: An Empirically Study

## Abstract

Reinforcement Learning from Human Feedback (RLHF) enhances the alignment between humans and large language models (LLMs), with Reward Models (RMs) playing a pivotal role. RLHF and sampling techniques, such as Best-of-N, require RMs to provide reliable rewards to guide policy training or sample selection. However, despite the advancement of LLMs, critical issues in RMs persist, such as overestimation on out-of-distribution (OOD) data (also known as reward hacking) and a preference for verbose outputs (length bias). These issues undermine the reliability of RM-generated rewards. Training an unbiased RM requires addressing these challenges, yet there is a lack of in-depth analysis on RMs. In this paper, we first decompose the RM training pipeline and identify three key aspects critical for developing an unbiased RM: 1) model architectures, 2) training paradigms, and 3) the influence of preference data. For each aspect, we conduct thorough empirical studies, revealing several insightful design considerations. Building on our findings, we develop an RM capable of mitigating the identified issues. This study represents the first comprehensive examination of various challenges from a holistic perspective in RM training, offering in-depth analyses of essential concerns and providing guidance for training unbiased RMs that can accurately guide downstream policies. The relevant code and models will be made publicly available.

## 1 Introduction

Reinforcement Learning from Human Feedback (RLHF) is attracting increasing research interest with the rapid advancement of large language models (LLMs), enabling better alignment of generated texts with human preferences (Ouyang et al., 2022; Dubey et al., 2024; Yang et al., 2024; Achiam et al., 2023). The standard RLHF process can be divided into two stages. First, a reward model (RM) is trained using collected preference datasets. Second, reinforcement learning (RL) policies are trained using the RM as a proxy to maximize rewards. In addition to RLHF, RMs can be employed in offline sampling techniques to select high-quality samples (Liu et al., 2023a; Dong et al., 2023). Despite the critical role RMs play in the LLM era, their underlying mechanisms and existing issues remain relatively underexplored.

The RM in large language models (LLMs) typically follows the Bradley-Terry assumption Bradley & Terry (1952), serving as a proxy to approximate oracle preferences. Conventionally, training an RM relies on preference data collected by humans or strong AI models Hu et al. (2024). However, previous studies Wang et al. (2024); Ramé et al. (2024); Quan (2024) have revealed that ambiguous preferences frequently occur within the data. For instance, Quan (2024) shows that agreement between pairwise preference data is only around 60%-70%, which hampers the model's ability to accurately learn preferences. Additionally, RMs struggle to generalize effectively to unseen scenarios. Even when sufficient offline preference datasets are available, a distribution gap between training and validation test sets persists. This gap compels RMs to assign higher rewards to previously unseen samples, leading to the reward-hacking problem (Skalse et al., 2022). Another critical issue with RMs is the length bias problem, stemming from Goodhart's law (Karwowski et al., 2023). After RLHF, LLMs often suffer from verbosity, as they tend to generate longer responses to achieve higher rewards (Singhal et al., 2023; Park et al., 2024). In such cases, both standard RMs and advanced models like GPT-4 (LLM-as-Judge) (Zheng et al., 2023) may be misled into assigning higher rewards due to their preference for length. These issues, whether originating from the inherent characteristics of RMs or the improper collection of preference data, pose significant challenges to developing unbiased RMs.

In this paper, we address the aforementioned issues of RMs and conduct the first comprehensive examination of training an unbiased RM. To achieve this, we begin with a thorough review of related works on RMs and revisit the RM training pipeline. From this analysis, we identify three critical influencing factors. **First, model architectures**. Recent studies have shifted from standard single RM training to using multiple experts or coun-

terparts (Quan, 2024; Eisenstein et al., 2023; Ramé et al., 2024; Coste et al., 2023). One notable approach is the adoption of mixture-of-experts (MoEs) (Quan, 2024), which mitigates bias through the use of multiple experts. Additionally, some studies employ multiple RMs by either ensembling their outputs (Eisenstein et al., 2023; Coste et al., 2023) or recomposing the parameter space (Ramé et al., 2024). We reexamine RMs using these architectures, offering insights on when to adopt them. **Second, training paradigms.** RM training generally utilizes ranking functions to align the RM with preferred responses while diverging from rejected ones. However, tuning LLMs with large datasets requires careful parameter engineering (Dubey et al., 2024), and the varying preference scales among pairwise data can lead to unstable training. We identify that the pairwise ranking problem is analogous to a multi-objective contrastive learning problem (Chen et al., 2020), and we discuss the effects of different training paradigms. **Third, the effect of preference data.** While the scaling laws of LLMs have been widely studied (Ouyang et al., 2022; Isik et al., 2024), only a few works have preliminarily explored how increasing the volume of preference data can improve overall performance (Touvron et al., 2023), or examined the impact of data quality (Wang et al., 2024). No comprehensive study has been conducted on the specific impact of preference data on RMs. We explore this influence from two perspectives: the noise within the training set and the length of the preference data. These three aspects form the basis of our discussion on training an unbiased RM.

Derived from the three-fold discussions, we propose a RM that achieves higher classification performance while providing neutral rewards for downstream tasks. As an early study in identifying crucial issues, we deconstruct the different components of RMs and offer solutions to address these challenges, laying the foundation for training an unbiased RM. In summary, our contributions are as follows:

- We conduct a comprehensive examination of the existing RM training pipeline, identifying two critical issues: reward hacking and length bias.

- We perform a thorough decomposition of RM components, providing a fine-grained analysis of the impact of model architectures, training paradigms, and the effect of preference data, along with several constructive suggestions.

- Building on these insights, we develop an unbiased RM that yields promising performance, demonstrating effective mitigation of the identified issues.

As an empirical study on RMs, we emphasize that our primary goal is to investigate the influence of diverse factors rather than pursue the highest performance. Throughout the discussion, we provide guidance on when to employ specific model architectures, how to select appropriate training paradigms, and how to organize preference data effectively for RM training. Additionally, we show that the two critical problems—reward hacking and length bias—are mitigated.

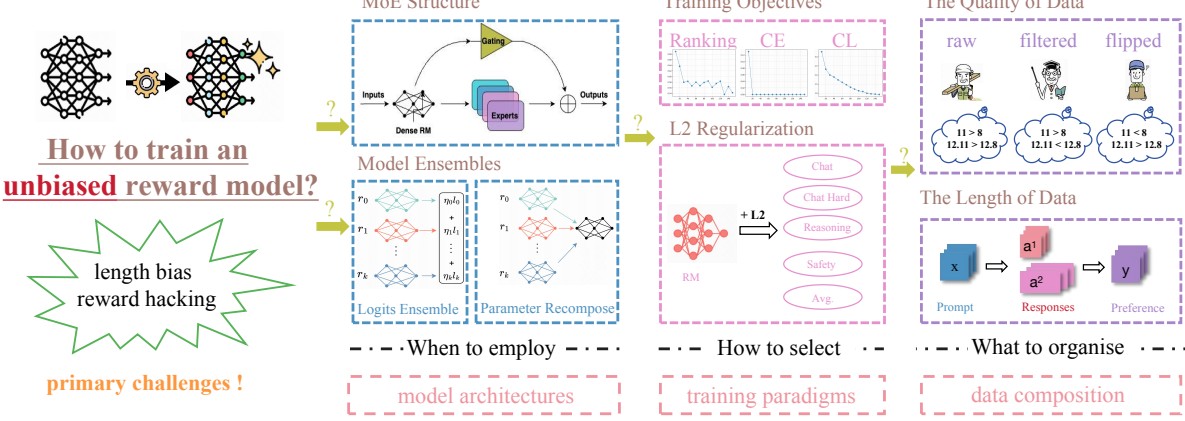

Figure 1: A simplified illustration of our work.

## 2 CONTEXT AND CHALLENGES

### 2.1 CONTEXT

**RM training pipeline.** RLHF requires explicit rewards for the generated samples; thus, a RM is introduced to act as a proxy to provide these rewards. A RM $r_\phi$ is a LLM parameterized by $\phi$, typically trained using the initial SFT model, $\pi_{SFT}$. The RM training process involves first collecting offline preference data, and then framing the training as a binary classification problem.

The preference data for RM training consists of a set of preference pairs, $\mathcal{D} = \left\{ x^{(i)}, y_w^{(i)}, y_l^{(i)} \right\}_{i=1}^N$, where $y_w$ and $y_l$ denote the preferred and dispreferred answers for a given input prompt $x$, respectively. The relative preference between $y_w$ and $y_l$ can be determined by human labelers or strong LLMs (Lee et al., 2023). Collecting comprehensive preference data is typically labor-intensive (Hu et al., 2024) and requires coverage across diverse categories to enhance generalization ability. Additionally, ambiguous data pairs can mislead the RM's optimization; thus, ensuring the diversity and proper preference order in pairwise preference data is crucial.

Based on the collected preference data, the RM is further trained as a binary classification problem. Generally, the RM's training objective aligns well with the assumption of the Bradley-Terry (BT) model (Bradley & Terry, 1952), which is well-known in preference learning (Ouyang et al., 2022; Dubey et al., 2024). The BT model specifies the preference distribution $p^*$ as:

$$p^* \left( y_1 \succ y_2 \mid x \right) = \frac{\exp \left( r^* \left( x, y_1 \right) \right)}{\exp \left( r^* \left( x, y_1 \right) \right) + \exp \left( r^* \left( x, y_2 \right) \right)}, \tag{1}$$

where $r^*$ represents the oracle RM the measures the rewards. Using the preference data $\mathcal{D}$, we can train a RM $r_\phi$ follow the log-likelihood loss as:

$$\mathcal{L}_R \left( r_\phi, \mathcal{D} \right) = -\mathbb{E}_{(x, y_w, y_l) \sim \mathcal{D}} \left[ \log \sigma \left( r_\phi \left( x, y_w \right) - r_\phi \left( x, y_l \right) \right) \right], \tag{2}$$

**RL fine-tuning.** Once the RM is obtained, we leverage it as a proxy to offer rewards for RL fine-tuning. Specifically, RL fine-tuning aims to maximize the following reward objective:

$$r_{\text{total}} = r_\phi(x, y) - \eta \text{KL} \left( \pi^{\text{RL}}(y \mid x) \| \pi^{\text{SFT}}(y \mid x) \right), \tag{3}$$

where $\eta$ is the coefficient that controls the magnitude of the KL penalty. Build upon Equation 3, RL algorithms like proximal policy optimization (PPO) (Schulman et al., 2017), direct preference optimization (DPO) (Rafailov et al., 2024) were developed.

Aside from its application in RL algorithms, the RM is also used to source good samples from the estimated target optimal policy using rejection sampling (Liu et al., 2023b). In this paper, apart from evaluating the RM with open benchmarks, we also test the RM with the Best-of-N (BoN) strategy, and further measure the quality of the chosen samples using alignment benchmarks.

### 2.2 CHALLENGES

The surge in LLMs has inspired increased research and real-world applications (Ouyang et al., 2022; Dubey et al., 2024; Yang et al., 2024). Over the past years, the research focusing on LLMs has shifted to post-training (Dubey et al., 2024), which heavily relies not only on providing reward signals for on-policy algorithms, such as PPO, but also on selecting refined responses in strategies like rejection sampling (Liu et al., 2023a). Therefore, several key challenges remain in the effective deployment and utilization of RMs.

**Length Bias** It is observed that the text generated after the RLHF stage tends to be longer (Singhal et al., 2023), while RMs are prone to assign higher rewards to longer sentences (Singhal et al., 2023; Dong et al., 2024). The longer text may contain redundant information compared to shorter, more concise text, which is infeasible for optimizing RL policies. As revealed in Singhal et al. (2023), simply using the length of training samples as a reward has brought significant improvements in downstream PPO tasks. Therefore, mitigating length bias in RMs is a huge challenge to overcome.

**Reward Hacking** Building a RM that can offer unbiased, accurate reward signals is non-trivial. The standard RM is trained on limited offline collected datasets, which inevitably brings distribution shift, resulting in the reward hacking problem (Skalse et al., 2022). In addition, as revealed in (Wang et al., 2024), the existing preference datasets are often noisy, containing incorrect and ambiguous preferences. This low-quality data issue further hinders the RM's ability to generalize to out-of-distribution (OOD) scenarios.

## 3 ANALYSIS OF RMs

After decomposing the RM training pipeline, we analyze the RM in this section, focusing on the effects of model structures in Subsection 3.1, training paradigms in Subsection Bai et al. (2022), and the influence of preference data in Subsection Hu et al. (2024).

Unless otherwise specified, all analyses are conducted based on the Llama3-8b-SFT model (denoted as $\pi_{sft}$) fine-tuned by Dong et al. (2024). To comprehensively explore the impact of data, we utilize preference data from two variants: the HH-helpful [1] dataset, which includes 115,396 preference data points (denoted as $\mathcal{D}_{Base}$), and the complete preference data collected by Dong et al. (2024) (denoted as $\mathcal{D}_{Full}$), which contains 1,090,979 preference data points, we list the full information of used datasets in Appendix A.1. We assess the RM's discriminative ability on RewardBench [2]. In addition, we assess the RM using the Best-of-N (BoN) technique. Specifically, based on the prompts from AlpacaEval 2.0 Li et al. (2023), we first use $\pi_{sft}$ to generate 64 samples for each prompt. We then employ the trained RM to select the sample with the highest score. Finally, we evaluate the selected sample according to the official evaluation procedure [3].

### 3.1 MODEL ARCHITECTURES

**MoE architecture**  We analyze the effect of the Mixture-of-Experts (MoE) architecture in this subsection. MoE architectures have garnered increasing interest in large language models (LLMs) Jiang et al. (2024); Dai et al. (2024) and yield promising results as sparse models. However, the application of MoE architecture to Reward Models (RMs) has been rarely investigated. An exception is Quan (2024), which proposes a double-layer MoE architecture that routes each input to corresponding task-specific experts, demonstrating significant improvements compared to vanilla RMs.

Our focus is to study the impact of the MoE architecture on RM learning. To reveal its influence, we conduct a straightforward comparative experiment. Based on the supervised fine-tuning policy $\pi_{sft}$, we train an RM, denoted as $r_\phi$, according to Equation 2, which is parameterized by $\phi$. To validate the impact of preference data size, we utilize two variants of preference data: $\mathcal{D}_{Base}$ and $\mathcal{D}_{Full}$ for RM training. The model $r_\phi$ is built upon the LLM with a linear layer that projects the output of the dense LLMs to a scalar value. We denote the single RM as $r_\phi^{single}$ and the RM with MoE as $r_\phi^{moe}$. For $r_\phi^{moe}$, we incorporate experts on top of $r_\phi^{single}$, along with a gating network to control the information that each expert passes through, as illustrated in Figure 2.

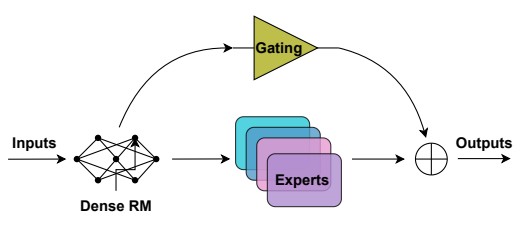

Figure 2: MoE Structure.

| Data | Models | Chat | Chat Hard | Reasoning | Safety | Avg. |
|------|--------|------|-----------|-----------|--------|------|
| Base | $r_\phi^{Base}$ | 93.6 | 66.0 | 82.7 | 46.8 | 72.6 |
|      | + MoE-2 | 94.7 | 66.0 | 82.9 | 48.0 | 73.1 |
|      | + MoE-4 | 93.6 | 66.5 | 83.7 | 45.5 | 72.8 |
|      | + MoE-6 | 94.7 | 65.4 | 84.5 | 47.8 | 73.7 |
|      | + MoE-8 | 94.7 | 65.1 | 88.0 | 52.4 | 76.5 |
| Full | $r_\phi^{Full}$ | 98.6 | 64.9 | 87.6 | 87.3 | 85.4 |
|      | + MoE-2 | 98.6 | 64.7 | 89.3 | 86.6 | 86.0 |
|      | + MoE-4 | 98.9 | 63.2 | 89.1 | 86.2 | 85.6 |
|      | + MoE-6 | 98.9 | 64.3 | 88.5 | 86.2 | 85.5 |
|      | + MoE-8 | 98.9 | 64.3 | 88.1 | 86.4 | 85.3 |

Table 1: Results of MoE experiments.

We develop four variants with different numbers of experts: 2, 4, 6, and 8. The results are reported in Table 1, from which several conclusions can be drawn. First, consistent with the findings of Dubey et al. (2024), an increase in data volume can significantly enhance overall performance. This is intuitive, as more data can alleviate the OOD phenomenon and facilitate generalization to more domains; for example, the performance in the safety domain nearly doubles. Second, we observe that the MoE architecture can better route the outputs of the dense RM under limited preference data, yielding generally higher performance than the vanilla RM. Additionally, we find that increasing the number of experts can further improve classification results, with the variant using 8 experts yielding the best performance, surpassing the base RM by 3.9%. However, no apparent benefits are

---

[1] https://huggingface.co/datasets/Anthropic/hh-rlhf

[2] https://huggingface.co/spaces/allenai/reward-bench

[3] https://tatsu-lab.github.io/alpaca$_e$val/

observed under the preference training data of $\mathcal{D}_{Full}$. This is likely because $\mathcal{D}_{Full}$ contains preference data covering various categories, leading each expert to learn similar knowledge after sufficient training. Furthermore, it is noteworthy that a small increase in parameter computation yields relatively significant performance improvements. Assuming that $k$ experts are adopted, each expert and the gating network occupies $k \times d$ parameters, resulting in a total parameter growth of $2k \times d$.

Therefore, under the reward-based Bradley-Terry model, in scenarios where sufficient preference data is lacking, reshaping the vanilla RM with an MoE architecture enhances RM performance, with the benefits being more pronounced when less preference data is available.

**Ensemble methods.** The RM ensemble aims to mitigate the bias of a single RM, thereby deriving a more robust RM by combining different RMs without the need for further training Coste et al. (2023). Several studies have discussed RM ensembles Eisenstein et al. (2023); Coste et al. (2023); Ramé et al. (2024); Zhang et al. (2024b), demonstrating their effectiveness. To summarize, the ensemble methods can be categorized into two types: logits ensemble (LE) and parameter recompose (PR). LE is used to combine the logits output by various RMs and estimates the ultimate reward using mean, worst-case, or uncertainty-weighted optimization Coste et al. (2023). Given $k$ RMs, LE requires $k$ times the inference. In contrast, PR is more lightweight. While PR also requires multiple RMs for input, it ensembles different RMs by recomposing their parameter space. Let $\phi_S = \{\phi_1, \phi_2, \ldots, \phi_k\}$ denote the parameter space of $k$ RMs. PR aims to recombine these parameters by averaging the parameter spaces, resulting in an ensembled parameter space, $\phi_{avg}$. During the inference stage, the model averaged by PR initializes with the averaged parameters, eliminating the overhead of loading multiple RMs and thus reducing inference costs.

To explore the explicit impact of the two ensemble methods, we conduct a comprehensive study based on multiple models. To ensure the diversity of the initial models and to investigate the influence of the number of ensembled models, we perform DPO under various experimental settings, resulting in four aligned models: Instruct-A, Instruct-B, Instruct-C, and Instruct-D. These four models, along with Llama3-8-SFT and Llama3-8-Instruct, form a diverse model zoo with distinct alignment capabilities on AlpacaEval 2.0. The win rates of these six aligned models satisfy the following order:

$$\text{Llama3-SFT} > \text{Instruct-A} > \text{Instruct-B} > \text{Instruct-C} > \text{Instruct-D} > \text{Llama3-Instruct}.$$

Without loss of generality, we conduct ensemble experiments using preference data from both $\mathcal{D}_{Base}$ and $\mathcal{D}_{Full}$. We present the detailed training procedures for the four DPO models, the explicit win rates of the six models, and the detailed ensemble results using $\mathcal{D}_{Full}$ in Appendix 7.

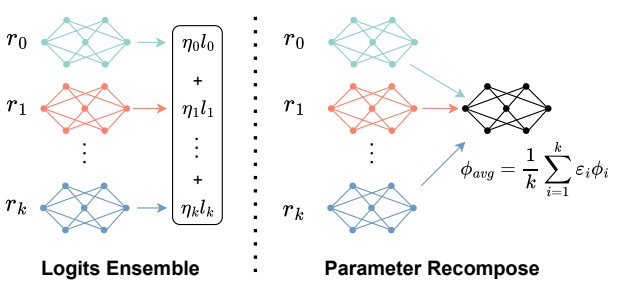

Figure 3: The two ensemble methods.

| | Models | Chat | Chat Hard | Reasoning | Safety | Avg. |
|---|---|---|---|---|---|---|
| | SFT | 95.5 | 63.6 | 85.7 | 42.3 | 72.8 |
| | Instruct | 93.6 | 66.0 | 82.7 | 46.8 | 72.6 |
| Single | Instruct-A | 95.8 | 64.5 | 88.5 | 43.2 | 74.5 |
| RM | Instruct-B | 95.3 | 63.8 | 83.4 | 43.4 | 71.9 |
| | Instruct-C | 95.3 | 63.4 | 83.7 | 44.7 | 72.3 |
| | Instruct-D | 94.1 | 63.8 | 78.6 | 44.1 | 69.7 |
| 2-Ens | LE | 94.7 | 66.2 | 86.7 | 44.3 | 74.0 |
| | PR | 94.7 | 66.7 | 79.5 | 45.8 | 71.0 |
| 4-Ens | LE | 95.8 | 65.8 | 87.8 | 43.8 | 74.5 |
| | PR | 94.7 | 58.6 | 79.5 | 48.4 | 70.4 |
| 6-Ens | LE | 95.3 | 64.9 | 86.4 | 44.1 | 73.7 |
| | PR | 95.6 | 58.1 | 79.4 | 47.8 | 70.3 |

Table 2: Results of ensemble experiments on $\mathcal{D}_{Base}$.

We conduct three groups of ensemble experiments with candidate RMs consisting of 2, 4, and 6 models. Specifically, we ensemble 2 RMs trained on Llama3-8B-SFT and Llama3-8B-Instruct, 4 RMs using Llama3-8B-SFT, Llama3-8B-Instruct, Instruct-A, and Instruct-B, along with an additional 2 RMs: Instruct-C and Instruct-D. The results are reported in Table 2 and Table 7, from which we can draw several observations. First, contrary to previous studies Ramé et al. (2024), model ensemble is not always beneficial; only the LE of RMs trained on Llama3-8B-SFT and Llama3-8B-Instruct using $\mathcal{D}_{Base}$ surpasses the performance of the base RMs. Second, PR generally deteriorates the performance of candidate RMs, which contradicts the conclusions made by Ramé et al. (2024), suggesting that PR can boost the overall win rate compared to the base SFT models. Finally, increasing the number of candidate RMs does not necessarily lead to higher performance. In addition, we also calculate the winrate of the two ensemble methods using RM trained on $\mathcal{D}_{Base}$ dataset and report the result in Appendix, both the two ensemble methods can facilitate select the better samples from the candidate pool.

## 3.2 TRAINING PARADIGMS

In this section, we aim to investigate different reward model training paradigms, focusing on the training objectives as well as the impact of L2 regularization.

**Training objectives**  Reward models commonly adhere to the Bradley-Terry model, utilizing the negative log-likelihood loss function, as illustrated in Equation2. Additionally, by assigning labels of 1 and 0 to preferred and non-preferred answers respectively, the preference modeling problem can be transformed into a binary classification task, thereby utilizing the cross-entropy (CE) loss function:

$$\mathcal{L}_{CE}(r_\phi, \mathcal{D}) = -\mathbb{E}_{(x, y_w, y_l) \sim \mathcal{D}} \left[ \log r_\phi(x, y_w) + \log(1 - r_\phi(x, y_l)) \right], \tag{4}$$

Wang et al. (2024) discovered that in reward modeling, the model often exhibits a high degree of feature similarity between preferred and dispreferred answers, making it challenging for the model to capture the subtle differences and distinctions between them. Introducing contrastive learning (CL) into the reward model can mitigate this issue and enhance the model's ability to discern these nuanced variations. SimCSE Gao et al. (2021) is a simple contrastive learning method for improving sentence representation:

$$\mathcal{L}_{CL}(r_\phi, \mathcal{D}) = -\mathbb{E}_{(x, y_w, y_{li}, \ldots, y_{lK}) \sim \mathcal{D}} \left[ \log \frac{e^{\text{sim}(\mathbf{h}_w, \mathbf{h}_w^+)/\tau}}{e^{\text{sim}(\mathbf{h}_w, \mathbf{h}_w^+)/\tau} + \sum_{j=1}^{K} e^{\text{sim}(\mathbf{h}_w, \mathbf{h}_{lj})/\tau}} \right], \tag{5}$$

where $(y_{li}, \ldots, y_{lK})$ are the $K$ dispreferred answers we collected as negative samples, $\mathbf{h}_w$ is the sentiment embedding from $(x, y_w)$, $\mathbf{h}_{lj}$ is the sentiment embedding from $(x, y_{lj})$, and $\tau$ denotes the temperature parameter. $\mathbf{h}_w^+$ is the sentence embedding of positive sample, which has the same input as $\mathbf{h}_w$ but with dropout. The final reward model loss is a combination of the native RM loss and the contrast learning loss: $\mathcal{L}_{total} = \mathcal{L}_R + \lambda \mathcal{L}_{CL}$, where $\lambda$ is a hyperparameter.

To investigate the distinctions among different training objectives, we trained the RMs on $\mathcal{D}_{Base}$ using three distinct loss functions: the original negative log-likelihood loss function (denoted as Ranking), the cross-entropy loss function (denoted as CE), and the contrastive learning loss function (denoted as CL). The base model employed was Llama3-8b-Instruct, and the evaluations were conducted on the reward bench. As shown in Table 3, the model trained with CL achieved the best results. This improvement can be attributed to the contrastive learning approach, which aids the model in distinguishing the subtle differences in representations between preferred and dispreferred answers. Consequently, it guides the model to effectively learn which outcomes are preferred. Conversely, the model trained with CE performed the worst. Figure 4 illustrates the training loss curves for models trained with the three different loss functions. From the figure, it is evident that the CE loss converges to a very low value within 20 iterations, suggesting that training with cross-entropy loss is overly simplistic for preference learning in reward models, leading to ineffective learning. The Ranking loss gradually converges to a smaller value, but the training process is unstable. In contrast, the contrastive learning loss demonstrates stable convergence throughout the training process.

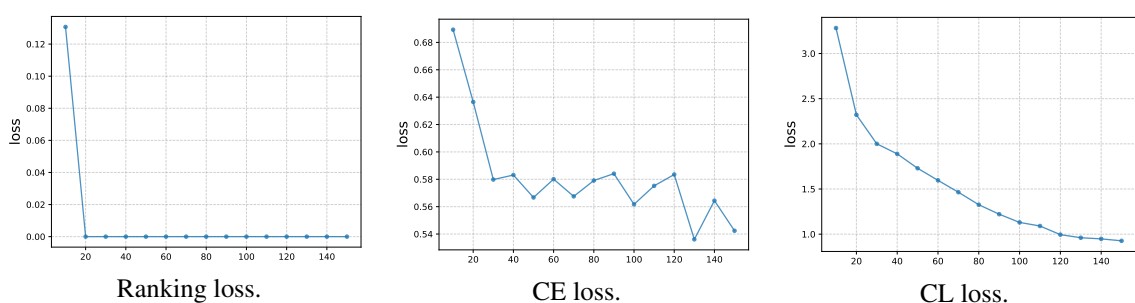

|  |  |  |
|:--:|:--:|:--:|
| Ranking loss. | CE loss. | CL loss. |

Figure 4: Training loss.

**Impact of L2 regularization**  L2 regularization is a common technique used to prevent overfitting by adding a penalty term to the loss function. This penalty term is defined as the expectation of the squares of response rewards:

$$\mathcal{L}_{L2}(r_\phi, \mathcal{D}) = -\mathbb{E}_{(x, y_w, y_l) \sim \mathcal{D}} \frac{1}{2} \left( r_\phi(x, y_w)^2 + r_\phi(x, y_l)^2 \right). \tag{6}$$

Table 3: Results of various losses and L2 regularization.

|      | Chat | Chat Hard | Reasoning | Safety | Avg. |
|------|------|-----------|-----------|--------|------|
| Raw  | 93.6 | 66.0      | 82.7      | 46.8   | 72.6 |
| +L2  | 93.6 | 68.2      | 85.7      | 48.4   | 74.7 |
| CE   | 82.1 | 56.4      | 63.7      | 46.5   | 60.5 |
| +L2  | 43.9 | 42.5      | 46.4      | 38.9   | 43.7 |
| CL   | 97.8 | 61.4      | 86.2      | 46.4   | 73.9 |
| +L2  | 96.7 | 62.7      | 87.8      | 46.6   | 74.8 |

We add the L2 regularization term to the original training Loss: $\mathcal{L}_{total} = \mathcal{L}_R + \beta\mathcal{L}_{L2}$, where $\beta$ is a hyperparameter.

To evaluate the impact of L2 regularization, we combined it with the three training objectives and trained the RMs, with the experimental results presented in Table 3. For both Ranking and CL, the inclusion of the L2 regularization term resulted in performance improvements. Notably, for the Ranking loss, the L2 regularization term contributed to a two-percentage-point performance increase, achieving results comparable to those obtained with the CL loss. As illustrated in the training loss curves in Appendix A.3, the L2 regularization term effectively stabilizes the training process. However, for CE, the addition of the L2 regularization term led to a performance decline. We hypothesize that this is because the CE training objective aims to score preferred answers as 1 and dispreferred answers as 0, while the L2 regularization term works to prevent the scores from deviating too far from 0. This inherent conflict between the CE objective and the L2 regularization may account for the observed performance degradation.

### 3.3 DATA EFFECT

In this subsection, we discuss the impact of data composition from two perspectives: 1) how to facilitate the RM training through data composition? and 2) how to mitigate length bias from the data aspect ?

**Noise study.** Previous research has indicated that the preference order in existing preference data may be unreliable due to label noise (Wang et al., 2024; Ramé et al., 2024; Coste et al., 2023). To investigate this, we conduct a progressive study on $\mathcal{D}_{Base}$ and $\mathcal{D}_{Full}$.

Take $\mathcal{D}_{Base}$ for illustration. Firstly, we leverage the ArmoRM (Dong et al., 2024) to score the pairwise preference samples of $\mathcal{D}_{Base}$. Secondly, we filter the samples that the rejected sample score higher than the chosen sample according the RM reward, resulting in the filtered dataset, $\mathcal{D}_{Base}^{Filter}$. This step filters out approximately 35% preference data, validating the presence of contradictions in $\mathcal{D}_{Base}$. Third, we step further by flipping the labels of the filtered samples, i.e., the chosen sample in $\mathcal{D}_{Base}$ is treated as the rejected sample, and vice versa. Then, we add the flipped samples back to the remaining samples in second step, forming $\mathcal{D}_{Base}^{Flip}$. It is worth noting that $\mathcal{D}_{Base}^{Flip}$ shares identical data volume with $\mathcal{D}_{Base}$, with labels of 35% of the preference data reversed. The identical operations are conducted on $\mathcal{D}_{Full}$.

Then, Based on $\mathcal{D}_{Base}^{Filter}$ and $\mathcal{D}_{Base}^{Flip}$, we train two RMs, namely, $r_\phi^{Base}$ and $r_\phi^{Full}$. Figure 5 presents the performance of $r_\phi$, $r_\phi^{Base}$ and $r_\phi^{Full}$. We observe that both the "filtering" and "flipping" operations facilitate model training. For $r_\phi^{Base}$, the filtering operation substantially improves the base performance by 11.1%, validating that removing noise from the preference dataset facilitates model learning. Besides, by simply reversing the filtered sample, $r_\phi^{Base}$ yields improvements of 2.1% and 13.2% compared to the RM trained on the vanilla and filtered preference samples, respectively. For $r_\phi^{Full}$, the improvements brought by the two operations is less resilient, while the "filtering" and "flipping" operations still yield improvements of 1.4% and 4.8% compared to the RM trained on the vanilla full preference data. Though relying on the external RM for the ranking, this consistent improvements in performance validate that the contradictory samples exist in $\mathcal{D}_{Base}$, suggesting that the noisy preference can be filtered, or reused by flipping the corresponding labels.

**Length bias.** As discussed in Section 1, RM is prone to be affected by the length of candidate samples, previous studies mitigate this issue by either adding constraint on the training objective Park et al. (2024) or restrain the difference in length of the pairwise responses Dong et al. (2024). Here, we explore the influence brought by the

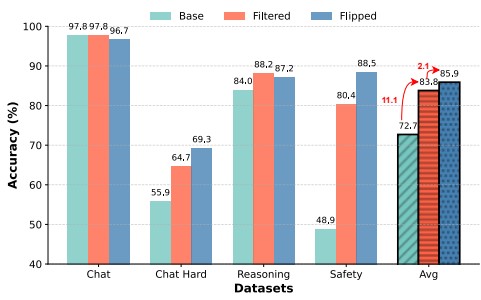 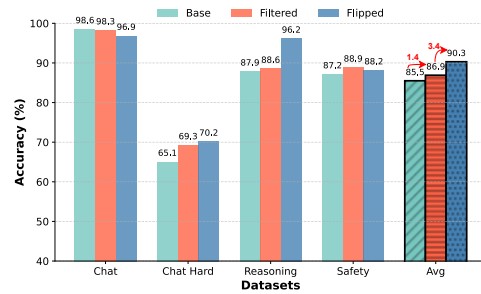

(a) The performance change of RM trained on $\mathcal{D}_{Base}$.    (b) The performance change of RM trained on $\mathcal{D}_{Full}$.

Figure 5: The performance change of $r_\phi^{Filter}$ and $r_\phi^{Flip}$.

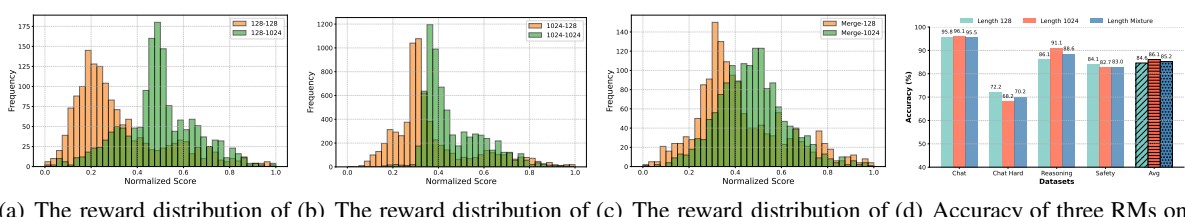

(a) The reward distribution of $r_\phi^{128}$.   (b) The reward distribution of $r_\phi^{1024}$.   (c) The reward distribution of $r_\phi^{mixture}$.   (d) Accuracy of three RMs on RewardBench.

Figure 6: The studies of length bias.

length of the training samples, which contains two steps: 1) the preference data construction of different length. 2) Training RM based on the constructed datasets. We using the prompts from $\mathcal{D}_{Base}$ to conduct experiments.

The first step entails construct two sets of preference data of different lengths, namely, 128 and 1024, respectively. To achieve this, we request the GPT4-turbo [4] to generate 8 diverse answers for each input, resulting in 8 responses, the specific instruction we used is presented in Appendix 8. In addition, we follow the same procedure to generate two testsets of length 128 and 1024, using the prompts from AlpacaEval 2.0 Li et al. (2023), denoted as $\mathcal{D}_{test}^{128}$ and $\mathcal{D}_{test}^{1024}$, respectively. Based on the generated 8 responses, we use ArmoRM Dong et al. (2024) to rank the 8 responses, with the response ranked highest as the chosen sample and the response ranked lowest as the rejected sample. Using the two preference datasets, we train two RM, denoted as $r_\phi^{128}$ and $r_\phi^{1024}$, respectively. We then leverage the $r_\phi^{128}$ and $r_\phi^{1024}$ to score the two testsets and plot the reward distribution in Figure 6(a) and Figure 6(b), respectively.

We can observe that both $r_\phi^{128}$ and $r_\phi^{1024}$ exhibit preference for length, regardless of the preference data they trained on. Our purpose is to mitigate the length bias from the perspective of data, which we wonder whether the mixture of preference data of diverse length can achieve. We validate this by combining $\mathcal{D}_{test}^{128}$ and $\mathcal{D}_{test}^{1024}$ into $\mathcal{D}_{test}^{mixture}$, then train a new RM $r_\phi^{mixture}$ using it. Analogously, we let $r_\phi^{mixture}$ to score the two testsets and plot the reward distribution in Figure 6(c). Compared to Figure 6(a) and Figure 6(b), $r_\phi^{mixture}$ exhibit much less salient dependence on sample's length, the two reward distributions tend to harmonize close to each other, successfully mitigating the reliance of RM scoring on length. Interestingly, the performance of the three RMs on RewardBench exhibits similar "harmonizing" phenomenon, $r_\phi^{mixture}$ exhibits mediate performance with $r_\phi^{128}$ and $r_\phi^{1024}$.

### 3.4 THE PROPOSED UNBIASED RM

Based on the comprehensive analysis above, we employed the ranking loss combined with L2 regularization as the loss function for training. Specifically, we filtered data of varying lengths from $\mathcal{D}_{Full}^{Flip}$, resulting in two datasets: $\mathcal{D}_{Full-short}^{Flip}$ and $\mathcal{D}_{Full-long}^{Flip}$. These datasets were used to train two separate RMs, denoted as M11 and

---

[4]https://platform.openai.com/docs/models/gpt-4-and-gpt-4-turbo

M12, respectively. Finally, we utilized the logits ensemble method to obtain the ultimate reward model, referred to as URM. The results of the our method are presented in Table 4 and Figure 7.

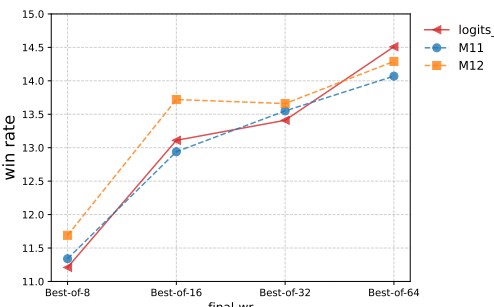 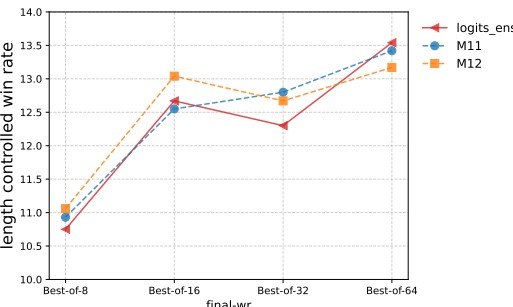

(a) Win rate of the proposed RMs on AlpacaEval 2.0.    (b) Length controled win rate of the proposed RMs on AlpacaEval 2.0.

Figure 7: Results on AlpacaEval 2.0

Table 4: Results on RewardBench

|      | Chat | Chat Hard | Reasoning | Safety | Avg. |
|------|------|-----------|-----------|--------|------|
| M11  | 97.0 | 70.2      | 96.2      | 88.2   | 90.3 |
| M12  | 95.8 | 73.3      | 96.7      | 87.6   | 90.8 |
| URM  | 96.7 | 74.3      | 96.9      | 88.9   | 91.4 |

## 4 RELATED WORKS

Reward models are essential components that leverage human feedback to fine-tune LLMs, thereby improving alignment with human preferences. Currently, RMs face four main challenges: length bias (Singhal et al., 2023), reward hacking (Skalse et al., 2022), distribution shifts, and inconsistent human preferences. Addressing these challenges is a key focus of ongoing RM research. This section reviews related work on RMs in three areas: model structures, training paradigms, and data composition.

### 4.1 MODEL STRUCTURES

To address the instability and potential biases in single reward model, two common architectures are employed: the MoE architecture (Quan, 2024) and model ensembling. These approaches also help mitigate issues such as reward hacking and misalignment with human intentions. DMoERM(Quan, 2024), leverages the MoE architecture to resolve issues related to multi-task interference and data labeling noise. This approach enhances generalization ability of the RM. In addition, numerous studies have focused on utilizing model integration to improving the robustness and reliability of RMs. For instance, Eisenstein et al. (2023) analyze the problem of prediction inaccuracies in RMs due to limited training data and examines methods for achieving more robust reward estimations through integrating multiple reward models. Coste et al. (2023) found that conservative optimization methods, such as Worst-Case Optimization (WCO) and Uncertainty-Weighted Optimization (UWO), effectively reduce over-optimization. Furthermore, Ramé et al. (2024) proposed Weight Averaged Reward Models (WARM). By fine-tuning multiple RMs and averaging them in the weight space, the reliability and robustness of the model can be enhanced under conditions of distributional shifts and preference inconsistencies. Zhang et al. (2024a) proposed two efficient reward model integration methods: linear layer integration and Low-Rank Adaptation (LoRA) basis integration, which aim to address prediction inaccuracies caused by limited training data.

### 4.2 TRAINING PARADIGMS

According to (Bai et al., 2022), the training of RMs can be broadly categorized into two paradigms: contrastive training and regression training. In contrastive training, RMs typically rely on the Bradley-Terry model to pull representations of positive pairs closer and push representations of negative pairs further apart. This allows

the RM to learn representations that capture key features of the data without the need for external annotation. Regression training refers to a supervised learning process utilizing regression models to predict continuous values. The regression algorithm attempts to find the relationship between input variables (features) and output variables (targets), focusing on minimizing the error between predicted and actual values. Additionally, due to the underdetermined nature of the Bradley-Terry model, which lacks a unique solution, integrating multiple reward models can present challenges. To address this, Eisenstein et al. (2023) added a regularization term to the maximum likelihood objective function, which ensures that the sum of the reward predictions for each preference pair trends towards zero Recently, Yuan et al. (2024) proposed a Self-Rewarding Language Model (SRLM), which provides its own rewards during training through a "LLM-as-a-Judge" prompt. By iterative DPO training, the model improves not only its ability to follow instructions but also its capacity to generate high-quality rewards.

### 4.3 Data Composition

Data composition has a significant impact on the training of RMs. Factors such as data distribution, quality, diversity, and balance all influence the effectiveness of RM training. Among these, the length of training texts and the composition of positive and negative samples are particularly important. Singhal et al. (2023) found that RMs inherently exhibit a strong preference for longer responses during training. Even when controlling for similar output lengths, RMs tend to assign higher reward scores to longer responses. Addressing this, Wu et al. (2024) proposed a fine-grained RM training approach, which provides rewards at each stage of text generation, for example, at the sentence level, and incorporates various reward models related to different feedback types. The composition of positive and negative samples significantly impacts RM training in three main ways: the intensity of positive samples, the diversity of negative samples, and the balance between positive and negative samples.

## 5 Conclusion

In this paper, we conduct research on developing an unbiased RM. We identify two critical problems in RM: the reward-hacking phenomenon and the length bias problem. Based on these two problems, we then decompose the RM training pipeline and identify three main aspects that impact RM performance: the model architecture, training paradigm, and preference data. For each perspective, we conduct a thorough empirical study, revealing intrinsic factors of RM and offering insights on developing a holistic, unbiased RM. We will continue in conduct deeper studies in combining RM with RL policies.

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

# A APPENDIX

## A.1 DATASETS

**Preference datasets.** The preference datasets we used for this paper is analogous from Dong et al. (2024). We list the details of these datasets below in Table 5.

| Datasets | Num. of Comparisons | Avg. # Tokens in Prompt | Avg. # Tokens in Chosen Response | Avg. # Tokens in Rejected Response |
|---|---|---|---|---|
| HH-Helpful | 115,395 | 16.8 | 82.2 | 73.6 |
| CodeUltraFeedback | 50,155 | 163.8 | 427.6 | 400.6 |
| Standford SHP | 93,300 | 176.0 | 173.5 | 88.8 |
| HelpSteer | 37,130 | 535.8 | 116.4 | 89.3 |
| PKU-SafeRLHF | 26,873 | 16.5 | 70.4 | 74.6 |
| UltraFeedback | 340,024 | 156.3 | 279.5 | 211.1 |
| UltraInternet | 161,926 | 279.5 | 396.6 | 416.7 |
| Distilabel-Capybara | 14,810 | 50.3 | 348.4 | 401.9 |
| Distilabel-Orca | 6,925 | 148.3 | 165.4 | 260.5 |

Table 5: Data information

## A.2 MODEL ARCHITECTURES

**Ensemble methods**

| Models | WR | LC_WR |
|---|---|---|
| SFT | 6.7 | 13.4 |
| Instruct | 13.0 | 22.3 |
| Instruct-A | 8.4 | 16.1 |
| Instruct-B | 9.4 | 15.6 |
| Instruct-C | 11.8 | 18.7 |
| Instruct-D | 12.6 | 20.9 |

Table 6: Single model Performance

Table 7: Results of ensemble experiments on $\mathcal{D}_{Full}$.

| Models | | Chat | Chat Hard | Reasoning | Safety | Avg. |
|---|---|---|---|---|---|---|
| Single RM | SFT | 97.8 | 60.8 | 96.7 | 87.0 | 88.9 |
| | Instruct | 98.6 | 65.1 | 87.9 | 87.2 | 85.5 |
| | Instruct-A | 97.5 | 60.1 | 95.7 | 87.0 | 88.3 |
| | Instruct-B | 97.5 | 59.2 | 95.4 | 86.6 | 87.9 |
| | Instruct-C | 97.8 | 61.2 | 95.3 | 86.5 | 88.2 |
| | Instruct-D | 97.2 | 59.9 | 95.3 | 86.9 | 88.0 |
| 2-Ensemble | LE | 98.3 | 63.2 | 95.5 | 87.0 | 88.8 |
| | PR | 98.0 | 65.4 | 93.7 | 86.5 | 88.1 |
| 4-Ensemble | LE | 98.0 | 61.2 | 96.4 | 87.3 | 87.3 |
| | PR | 98.3 | 58.8 | 93.4 | 85.7 | 86.8 |
| 6-Ensemble | LE | 97.8 | 61.2 | 96.7 | 86.9 | 88.9 |
| | PR | 98.3 | 57.5 | 92.0 | 85.3 | 85.8 |

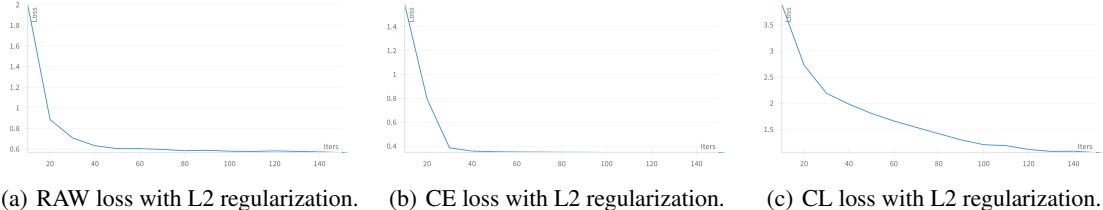

(a) RAW loss with L2 regularization.   (b) CE loss with L2 regularization.   (c) CL loss with L2 regularization.

Figure 8: The training process with L2 regularization

### A.3 TRAINING PARADIGMS

The training process shown in Figure 8

### A.4 DATA EFFECTS

### A.5 EXPERIMENT SETTINGS

**Implementation Details.**    The training parameter settings for the reward models are shown in Table 9.

### A.6 BEST-OF-N EXPERIMENT RESULTS

**Influence of MoE on model performance.**   Based on the 110,000 data of HH-Helpful and the 1.09 million data of RLHFFlow, we made a series of comparative experiments on the base model and the model with 2, 4, 6 and 8 layers of MoE architecture. The win rate and length controlled win rate measured on Alpaca-Eval are shown in the following figure.

**Influence of model ensemble on model performance.**   Based on the 110,000 data of HH-Helpful, we made an experiment to explore the influence of model ensembles. We experiment with two ensemble methods: logits

| Request Instruction | |
|---|---|
| Length 128 | Please provide a detailed response to the following question, ensuring that the answer is approximately 128 words in length.\nquestion: {data['instruction']})\nanswer: |
| Length 1024 | Please provide a detailed response to the following question, ensuring that the answer is approximately 1024 words in length.\nquestion: {data['instruction']})\nanswer: |

Table 8: Request Instruction

Table 9: Training parameter settings

| Param. | Value |
|---|---|
| learning rate | 2e-6 |
| weight decay | 0.001 |
| training epoch | 1 |
| batch size | 32 (16 for contrasting learning methods) |
| max length | 512 |
| gradient accumulation steps | 2 |

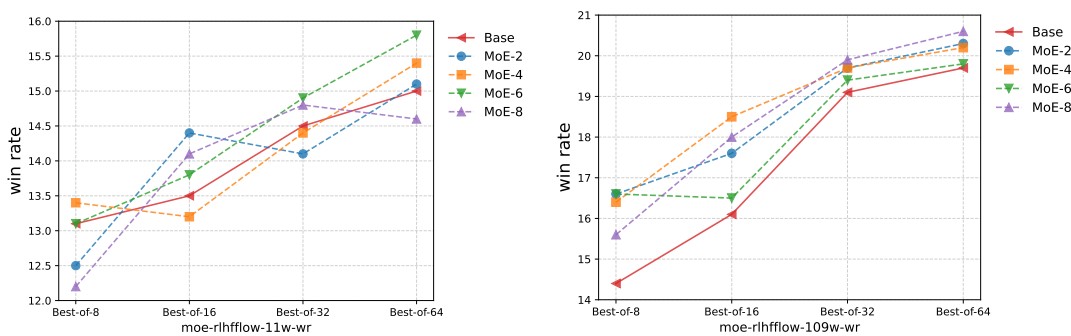

(a) Win rate of the RM trained on $\mathcal{D}_{Base}$ on AlpacaEval 2.0.

(b) Win rate of the RM trained on $\mathcal{D}_{Full}$ on AlpacaEval 2.0.

Figure 9: Win rate of the RM trained using different preference datasets on AlpacaEval 2.0.

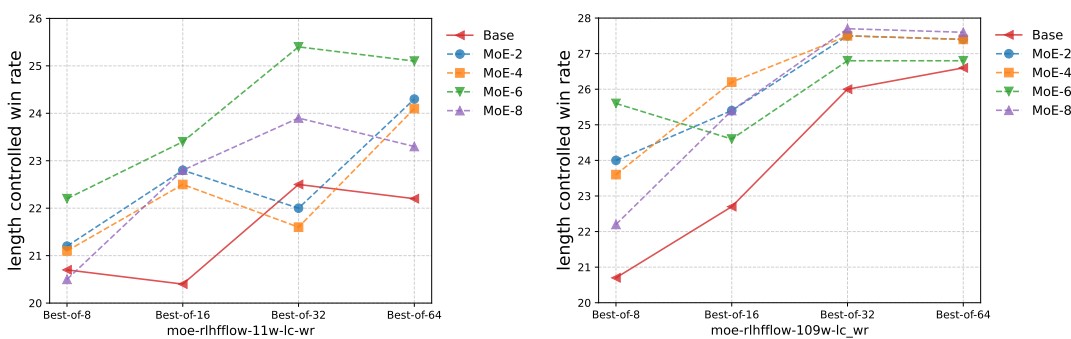

(a) Length controlled Win rate of the RM trained on $\mathcal{D}_{Base}$ on AlpacaEval 2.0.

(b) Length controlled Win rate of the RM trained on $\mathcal{D}_{Full}$ on AlpacaEval 2.0.

Figure 10: Length controled win rate of the RM trained using different preference datasets on AlpacaEval 2.0.

ensemble and parameter recompose. The win rate and length controlled win rate measured on Alpaca-Eval are shown in the following figure.

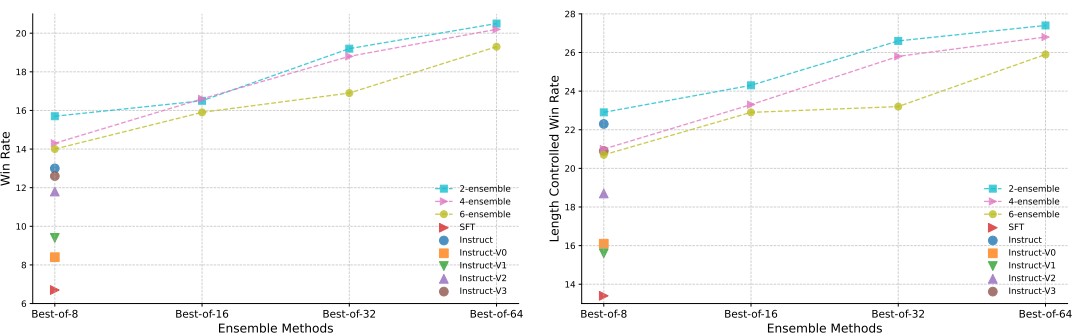

(a) Win rate experiment results using logits ensemble.

(b) Length controlled win rate using logit ensemble.

Figure 11: Performance of RMs using LE.

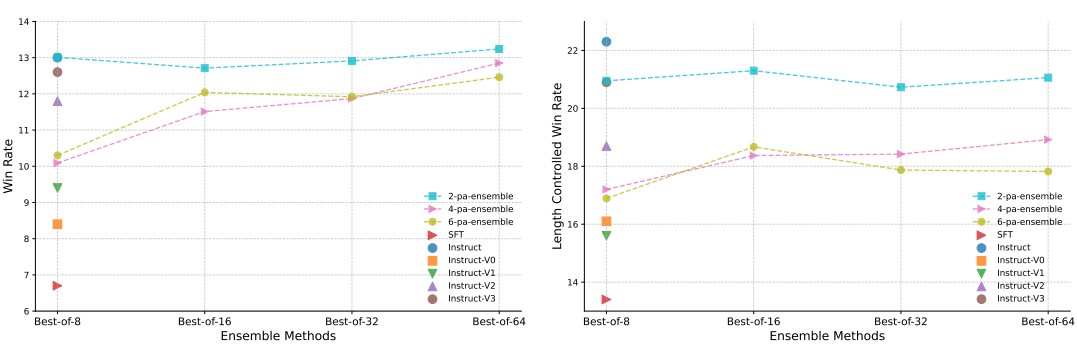

(a) Win rate experiment results using PR.

(b) Length controlled win rate experiment results using PR.

Figure 12: Performance of RMs using PR.

