# OpenReview forum: "Recipes for Unbiased Reward Modeling Learning: An Empirically Study"
_ICLR.cc/2025/Conference — ICLR 2025 Conference Withdrawn Submission_

### Official Review · Reviewer_7zc7 · 2024-10-17

**Soundness:** 2
**Presentation:** 2
**Contribution:** 2
**Rating:** 3
**Confidence:** 4

**Summary:**

This paper aims to address 1) reward hacking and 2) length bias in reward model training for LLM alignment.

To address the issues, this paper studies how the following three factors affect the reward model: 1) model architectures: dense model, MOE, logits ensemble, weight ensemble; 2) training paradigms: various loss functions with different regularization; 3) the influence of preference data: long, short, and mixed responses.

This paper chooses the best configuration of each factor and forms a training recipe for unbiased reward model.

**Strengths:**

To the best of my knowledge, this paper is the first to study how the model architecture, training paradigm, and preference data affect the reward model by evaluating the reward model on reward bench and win rate of the aligned LLM.

**Weaknesses:**

1. This paper does not study how the proposed three factors affect each other. It can be improved by examining whether these three factors are orthogonal to each other or if one of them is dominant.

2. The reward model ensemble experiments are confusing. The authors change the training objective from Eq. (2) (Line 132) to DPO, resulting in an unfair comparison. Furthermore, the ensemble of M11 and M12 in Section 3.4 is not consistent with the "Ensemble Methods" in Section 3.1. M11 and M12 are trained using Eq. (2), while the models in "Ensemble Methods" are trained with DPO.

3. The authors claim to address reward hacking but fail to demonstrate the relationship between the proxy reward (i.e., the reward from the reward model) and the true reward. If the true reward is not available, the outcome of a stronger reward model is also acceptable.

Minor presentation issues:
* Please use a regular font (i.e., \rm{}) if the superscript and subscript contain text. For example, D_{\rm{full}}.
* Incorrect citations on Lines 165 and 166.
* Please check for grammatical errors, such as those on Line 137 and Line 406.
* Please provide details for all the tables and charts. Readers should understand the tables and charts without referring to the main paper.

**Questions:**

1. At Line 186. What do you mean by “Based on the supervised fine-tuning policy $\pi_{\rm{sft}}$”? Do you mean the pari-wise data are sampled from the SFT model, or the RM model is initialized from the SFT model?

2. How do you obtain Instruct-A, Instruct-B, Instruct-C, and Instruct-D?

3. The overlap in Figure 6(c) looks more than Figure 6(a) and Figure 6(b). However, the range of y-axis in Figure 6(b) is much higher than Figure 6(c). Please change it to dense (normalized along y-axis) instead of frequency. Does your conclusion still hold?

4. How many parallel training are taken? What is the variance of your models on reward bench? Is the improvement significant?

---

### Official Review · Reviewer_TA7z · 2024-11-03

**Soundness:** 2
**Presentation:** 2
**Contribution:** 3
**Rating:** 5
**Confidence:** 5

**Summary:**

This paper seeks to train an unbiased reward model, such that the reward model is not easy to “hack” and does not have a length preference. To do so, they empirically study a variety of design choices, including model architecture, training paradigms, and preference data. As a final result, they show strong performance on reward bench.

**Strengths:**

- Studying how different design choices of reward modeling combine is an important area.
- The final result is quite a strong reward bench score.

**Weaknesses:**

- I think the main weakness of this paper is that the findings are not very novel. The authors show that by combining a few approaches from other papers can result in a better model, but this is kind of expensive.
- Another weakness of this paper is that it is not a very comprehensive empirical study. For MOE approaches, I would expect the authors to include the ARMO-RM approach (which has claimed good results) but they do not. Similarly, for the learning objective,  the authors do not include some interesting approaches like the generative RM approach.
- The writing is not that clear (see my questions).

**Questions:**

- What is “Raw” in table 3?
- I think some more clear descriptors than M11 and M12 should be used.
- On line 75, the authors say their contribution is “identifying two critical issues: reward hacking and length bias”. These two have certainly been identified before, maybe the authors could describe their contribution as “studying  reward hacking and length bias”?

---

### Official Review · Reviewer_b8SK · 2024-11-04

**Soundness:** 2
**Presentation:** 4
**Contribution:** 2
**Rating:** 5
**Confidence:** 2

**Summary:**

The authors develop a Unbiased Reward Model (URM) by ensembling tailored RMs, leading to classification performance improvement and lower verbosity preferences and overestimation of OOD rewards. It tests these results across a suite of RMs.

**Strengths:**

- Unbiased Reward Model (URM) gives actual applied guidance for addressing RM biases, it's 91.4% average score achieved in RewardBench.
- Logits ensemble & parameter recomposition a creative workaround to the instability of RM predictions, helping combat the tendency towards reward overestimation for OOD data.

**Weaknesses:**

- The work is lacking in data diversity exploration, e.g. how different categories of prompts (questions vs. statements? and semantic diversity of other types) affect RM generalization and performance.
- Parameter recomposition (PR) experiments resulted in performance deterioration, so it might not be generally effective.

**Questions:**

- Please provide a more detailed discussion on the boundary conditions for the MoE's utility. Answer, e.g. under what conditions should a human designer choose MoE or avoid PR?

---

### Official Review · Reviewer_HG6g · 2024-11-06

**Soundness:** 1
**Presentation:** 1
**Contribution:** 1
**Rating:** 1
**Confidence:** 5

**Summary:**

N/A

**Strengths:**

N/A

**Weaknesses:**

Upon reviewing the manuscript, I noticed that the authors have reduced the page margins on all sides. This modification does not align with the ICLR's formatting guidelines and this paper should be desk rejected.

**Questions:**

N/A

---

### Note · Authors · 2024-11-12

I have read and agree with the venue's withdrawal policy on behalf of myself and my co-authors.